# Crown Fire Modeling and Its Effect on Atmospheric Characteristics

Egor Loboda [1], Denis Kasymov [1,*], Mikhail Agafontsev [1], Vladimir Reyno [2], Anastasiya Lutsenko [1], Asya Staroseltseva [1], Vladislav Perminov [3], Pavel Martynov [1], Yuliya Loboda [1] and Konstantin Orlov [1]

[1] Department of Physical and Computational Mechanics, National Research Tomsk State University, 634050 Tomsk, Russia
[2] Laboratory of Wave Propagation, E. Zuev Institute of Atmospheric Optics of the Siberian Branch of the Russian Academy of Science, 634055 Tomsk, Russia
[3] Department of Forestry and Landscape Construction, Biological Institute, National Research Tomsk State University, 634050 Tomsk, Russia
* Correspondence: kdp@mail.tsu.ru

**Abstract:** The article is concerned with the experimental study of the crown fire effect on atmospheric transport processes: the formation of induced turbulence in the vicinity of the fire source and the transport of aerosol combustion products in the atmosphere surface layer at low altitudes. The studies were carried out in seminatural conditions on the reconstructed forest canopy. It was established that the structural characteristics of fluctuations of some atmosphere physical parameters in the case of a crown fire practically coincide with the obtained earlier values for a steppe fire. The highest concentration of aerosol combustion products was recorded at a height of 10–20 m from the ground surface. It was found that the largest number of aerosol particles formed during a crown fire had a particle diameter of 0.3 to 0.5 μm. As a result of experimental data extrapolation, it is concluded that an excess of aerosol concentration over the background value will be recorded at a distance of up to 2000 m for a given volume of burnt vegetation. It is of interest to further study these factors of the impact of wildfires on atmosphere under the conditions of a real large natural wildfire and determine the limiting distance of aerosol concentration excesses over background values.

**Keywords:** crown fire; aerosols; mass transfer; atmosphere

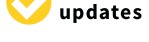



## 1. Introduction

Currently, landscape wildfires are one of the largest natural disasters that involve large adverse effects (air pollution, destruction of ecosystems and biodiversity, forest degradation, and economic losses). The predicted global climate change may lead to an increase in the frequency of forest fires, as well as the duration of the fire season, which will inevitably lead to an increase in the number of large and catastrophic wildfires, spread of their distribution area, and long-term degradation of forest conditions [1]. This affects the background radiation, cloudiness, air quality, and climate on a regional and global scale [2–4]. The composition and number of emissions from wildfires into the atmosphere depend on the characteristics of the combustible material, volume, structure, type, chemical composition, humidity, and fire behavior. Forest fires emit enormous amounts of gaseous components into the atmosphere and significantly affect the cycle and accumulation of carbon in boreal forests [5]. Furthermore, a huge amount of combustion products (gases and aerosols) is released into the atmosphere, which can be dispersed over vast distances [6–8]. For example, an excess of CO and $NO_x$ can change the oxidizing capacity of the atmosphere and significantly disrupt the background chemical composition of the atmosphere [9]. It has been proven that the concentration of such substances in the atmosphere can have a detrimental effect on air quality, health, and climate [10,11]. In particular, the negative contribution from excess CO, $SO_2$, and $NO_x$ was shown by performing a sensitivity analysis of the correlation between wildfires and carbon accumulation in living biomass, soil, and ground layer of boreal forests [12,13].

The record shows that a significant amount of thermal energy is released during combustion, and intense turbulent processes in the flame of a wildfire form turbulence in the convective column above the combustion source [14]. Obviously, this affects meteorological parameters: wind speed, induced atmospheric turbulence, changes in temperature, and relative air humidity [15–17]. It is known [18] that large forest fires form their "own wind" (induced wind), which in turn stimulates the wildfire spread and prevents it from termination. It should also be noted that massive natural wildfires are accompanied by stable anticyclone phenomena [18,19], which prevent the formation and fall of precipitation. Obviously, a massive release of thermal energy during large wildfires, accompanied by turbulent processes, affects the dynamics of atmospheric processes, and affects global climatic processes along with the release of carbon monoxide and small aerosols. On the other hand, changes in meteorological parameters directly affect the transfer of gaseous combustion products, plume, and aerosol. Changes in these parameters directly affect the transfer of gaseous combustion products, smoke, and aerosol. Research in this area is carried out using both experimental methods [17,19–22] and mathematical models [23–25].

Some numerical results published over the past decade using a fully physical approach are presented and discussed with emphasis on the model [26]. Numerical simulations are compared with experimental data obtained at various scales: from laboratory to field wildfires in pastures and boreal forests. Some perspectives are presented regarding the potential link between physical wildfire models and atmospheric models in order to study the effects of wildfires on a larger scale. Although such models help one to make fire management decisions, they do not take into account the interaction between wildfire and the environment (atmospheric turbulence caused by wildfire). Therefore, several researchers have turned to computational fluid dynamics (CFD) models to investigate the detailed flow dynamics underlying wildfire behavior [27].

Despite the diverse studies of the natural wildfires impact on the environment and the air state [1–3,5], there is still no understanding of the phenomenon complexity and theoretical models of the wildfires impact on global climatic processes associated with climate change both locally and globally. Obviously, the impact of natural wildfires on the climate is caused not only by changes in the landscape and biogeocenoses [10,11], but also by physical and chemical processes occurring in the fire area and atmosphere as a result of the significant amount release of energy, gaseous, and condensed combustion products.

Tomsk State University in collaboration with the Institute of Atmospheric Optics SB RAS carry out long-term studies of wildfires under various conditions, including ones that are close to natural at the experimental site [28–33]. As a result, significant experience and knowledge has been accumulated in organizing and conducting such studies, as well as obtaining data on the natural wildfire front characteristics, its occurrence and spread, the effect of a wildfire on meteorological parameters, the characteristics of turbulence in the atmosphere, emissions, and transfer of combustion products [34].

This article presents the results of seminatural experimental studies of the crown fire occurrence on the modeled forest canopy and its effect on the atmosphere characteristics: the formation of induced atmospheric turbulence and the transfer of aerosol combustion products in the surface layer of the atmosphere at low altitudes. The obtained experimental results broaden the fundamental knowledge about the effect of forest fires on formation of induced atmospheric turbulence and atmospheric transport processes in general.

## 2. Experiment Explanation and Used Equipment

Experiments on modeling of the crown fire occurrence were carried out on 30 April and 5 May 2022 on the territory of the Basic Experimental Complex (BEC) of the Institute of Atmospheric Optics SB RAS [22]. The dimensions of the experimental sites were 4 × 10 m. Figure 1 shows a satellite image of the BEC with the locations of the experimental sites and measuring equipment marked on it. It should be noted that the fundamental difference between this work and similar works [14,30–32] carried out at the BEC before is the simulation of crown fire on the reconstructed forest canopy.

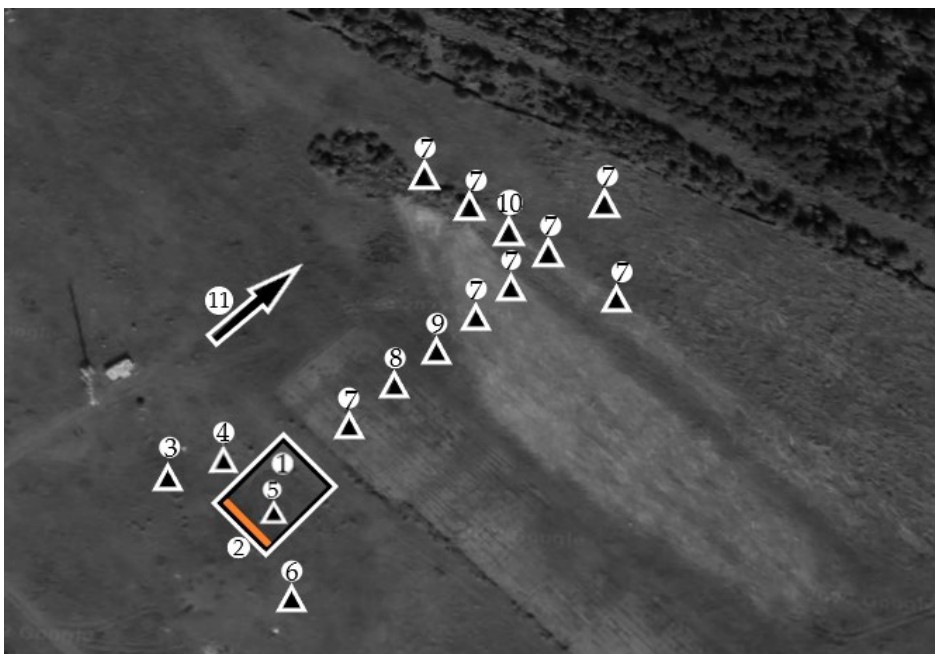

**Figure 1.** Satellite image of the BEC and the experimental site areas with measuring equipment: (1) experimental area, 10 × 4 m; (2) ignition line; (3) monitoring and recording equipment (PC and data logger "ZET LAB"); (4) JADE J530SB IR camera; (5) rack with thermocouples; (6) video camera; (7) rack with aerosol sensors; (8) AMK-03 weather station on a 3 m mast; (9) AMK-03 weather station on a 10 m mast; (10) AMK-03 weather station on a 6 m mast; (11) predominant wind direction during the experiment.

The air temperature, relative humidity, and atmospheric pressure were monitored using meteorological stations (AMK-03 ultrasonic weather station). Air temperature, T, varied within 275–278 K. Relative air humidity varied from 42% to 44%. Atmospheric pressure, Pe, was 713–730 mm Hg. The wind speed varied in the range of 1–6 m/s.

The moisture content of fuel vegetable materials (FVMs) was determined using an AND MX-50 moisture analyzer with an accuracy of 0.01% and equal to $W = 5.6\%$. The capacity of FVM on the experimental site varied within 0.476–0.563 kg/m$^2$. The temperature field in the wildfire front and the flame structure were monitored using a JADE J530SB infrared camera with a shooting rate of 50 frames/s in a narrow spectral range of 2.5–2.7 μm. The choice of the spectral interval is determined by the emission spectrum of the main combustion products of the flame [34]. Racks with CA (chromel–alumel) thermocouples type K were placed inside the experimental strip to correct the flame emissivity and control the wildfire front propagation [33]. The transfer of aerosol combustion products was controlled using a network of PMS 7003 ground-based sensors located at a height of 2 m at various distances (up to 105 m) from the combustion source. One sensor was attached to the unmanned aerial vehicle (UAV) and recorded the concentration of aerosol particles at a height of 20 m.

The experimental site (Figure 2) was an "accelerating site" of a ground fire, 1, an area of undergrowth and shrubs, 2, and a model forest canopy, 3. Zone 1 was ignited similarly [14] uniformly over the entire width (Figure 3). Zones 2 and 3 were reconstructed from undergrowth ($H_m = 1.2$ m) and pines ($H_s = 2.5$ m), which were preharvested during thinning in the territory of Tomsk forestry. Moisture content of vegetation was kept at natural values (moisture content of needles was $W = 114\%$). The maximum height of trees ($H_b$) in the reconstructed forest canopy did not exceed 4.5 m. The lengths of the sections were $h_{r1} = 2$ m; $h_{r2} = 2.5$ m; $h_{r3} = 3$ m.

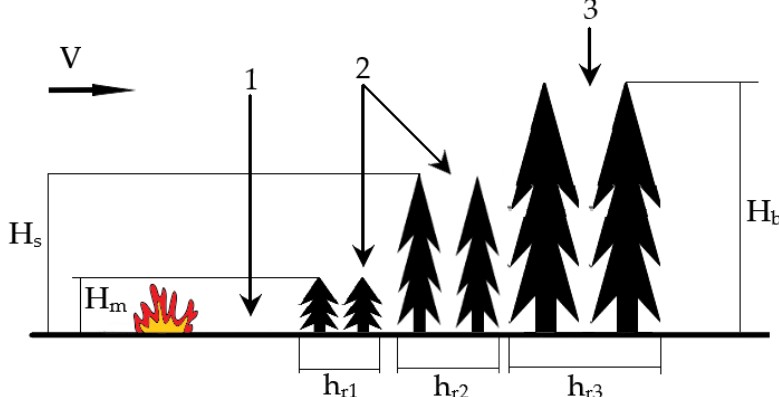

**Figure 2.** Scheme of the experimental site (side view): (1) "accelerating site" of a ground fire; (2) undergrowth and shrubs; (3) model forest canopy.

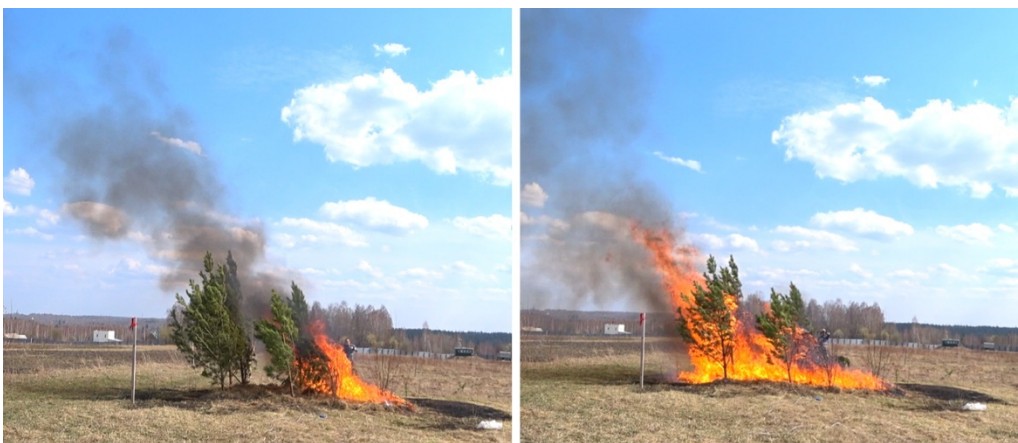

**Figure 3.** Image of the initial moment of a ground fire formation, and the transition of a ground fire to a crown fire.

### 3. Experimental Results and Analysis

The IR thermogram (Figure 4) shows the temperature distribution in the flame at various moments in time during the transition from a ground fire to a crown fire. Obviously, the combustion process is essentially unsteady and is accompanied by developed turbulence in the flame.

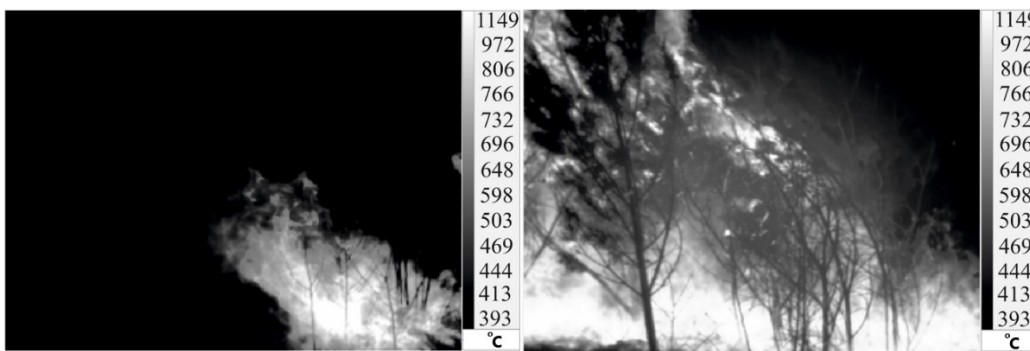

**Figure 4.** Flame thermogram of a model wildfire front.

Ref. [14] shows that turbulent processes in the flame lead to the formation of induced atmospheric turbulence as a result of dissipation and general heat release, which is shown in the value change of the structural constant of fluctuations of the refractive index, $C_n^2$,

obtained by optical and acoustic methods, as well as the structural constant fluctuations in temperature, $C_T^2$, and wind speed, $C_V^2$, which is associated with changes in air density. Figure 5 shows graphs of fluctuations in the refractive index, which were calculated using the following expressions:

$$C_T^2 = \left\langle \left[ T'(t + \Delta t) - T'(t) \right]^2 \right\rangle (\langle V_m \rangle \Delta t)^{-2/3} \tag{1}$$

$$C_{na}^2 = \frac{C_T^2}{(2\langle T_k \rangle)^2} + \frac{C_V^2}{\langle c \rangle^2} \tag{2}$$

$$C_{no}^2 = \left[ 8 \cdot 10^{-5} \frac{\langle P \rangle}{\langle T_k \rangle^2} \right]^2 C_T^2 \tag{3}$$

$$C_V^2 = \left\langle \left[ u'(t + \Delta t) - u'(t) \right]^2 \right\rangle (\langle V_m \rangle \Delta t)^{-2/3} \tag{4}$$

where $C_T^2$ is the structural constant of temperature fluctuations; $C_{na}^2$—structural constant of acoustic refractive index fluctuations; $C_{no}^2$—structural constant of fluctuations of the optical refractive index; $C_V^2$—structural constant of wind fluctuations; $T'$—value of turbulent temperature fluctuation, °C; $t$—current moment of time, s; $\Delta t$—time interval between measurements of instantaneous meteorological variables, s; $V_m$—module of the average wind speed vector, m/s; $<\ >$—symbol of statistical averaging; $T_k$—air temperature, K; $c$—speed of sound, m/s; $P$—atmospheric pressure, hPa; $u'$—value of the turbulent fluctuation of the wind speed component, m/s.

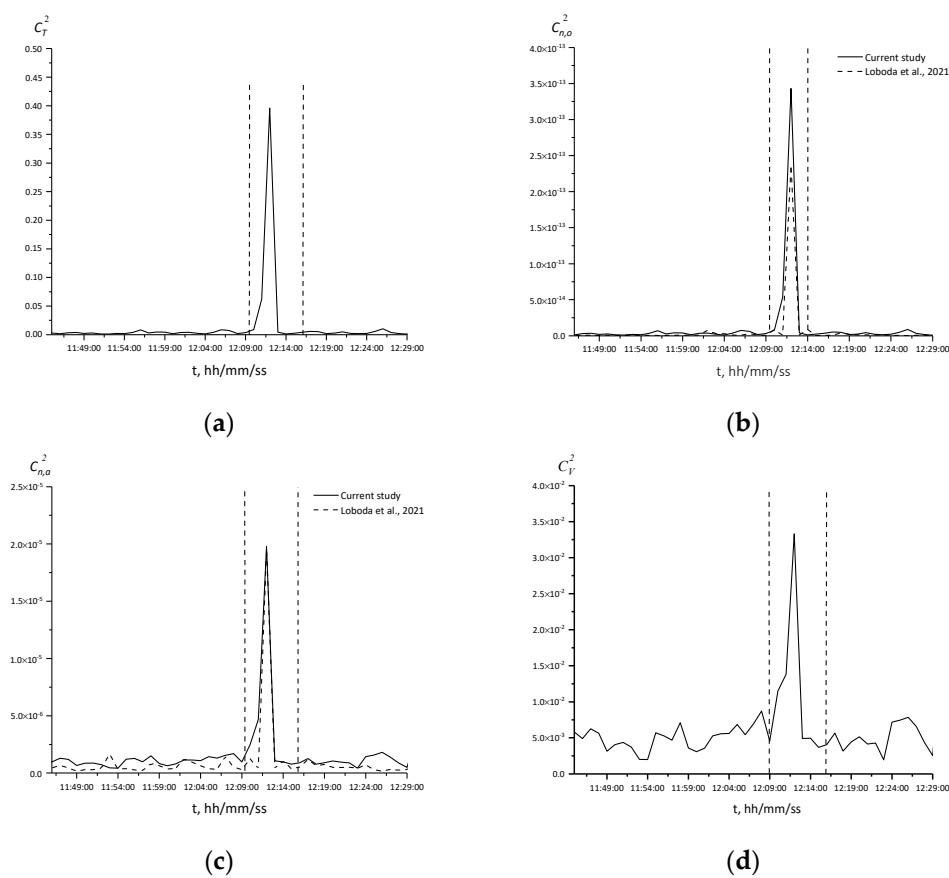

**Figure 5.** Change in the structural index of fluctuations in the refractive index of $C_n^2$, obtained by optical (**a**) and acoustic methods (**b**); structural index of fluctuations in wind speed (**c**) and temperature (**d**).

The magnitude of the structural characteristic of the air refractive index fluctuations is often used in problems of atmospheric optics for description of the turbulence effect on optical radiation. The work [35] shows justification for the fact that turbulence in the atmosphere surface layer has a decisive influence on the profile $C_n^2(z)$ up to heights of the order of several kilometers; $C_n^2(z)$ is the vertical profile of the structural characteristic of air refractive index fluctuations. Comparing the data shown in Figure 5 with the data published in [14], obtained from modeling of steppe fires, one can conclude that the value of $C_n^2$ in the cases of steppe and crown fires has the same order, which is associated with the same physical mechanisms of formation atmospheric turbulence—heat release in the combustion zone and dissipation of turbulent structures in the flame. Considering the fact that the characteristic dimensions of the combustion zone and the characteristic temperatures in the flame are similar for the experiment carried out and the experiment in [14], then, accordingly, one observes values of $C_n^2$, etc., similar in their values. It should be noted that, similarly to [14], one can register an increase in air temperature by 2–3 degrees and a change in wind speed.

It was found that the largest number of particles emitted into the atmosphere during a crown fire have a diameter of 0.3–0.5 μm (Figure 6) as a result of measurements of the aerosol particles concentration, which coincides with the data [14] obtained for a model steppe fire.

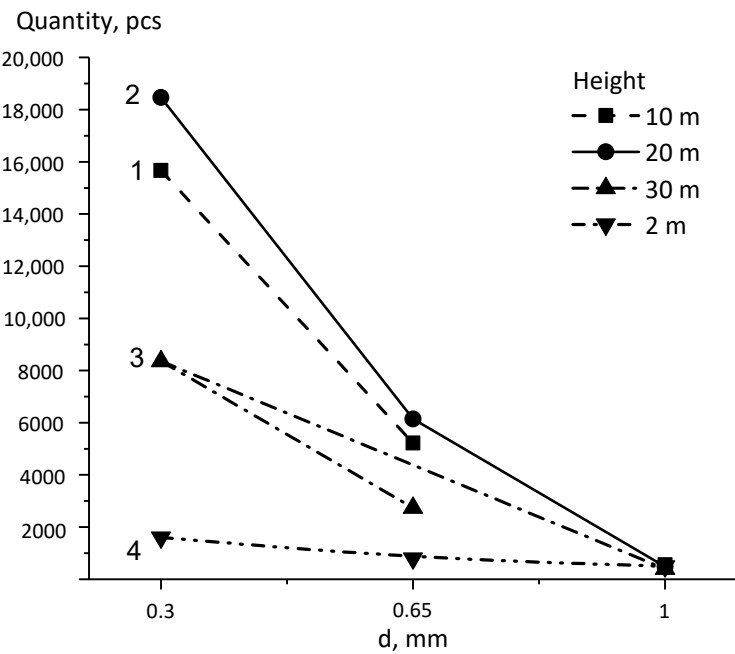

**Figure 6.** Distribution of aerosol concentration by particle diameter at a distance of 125 m from the combustion source at different heights.

Analyzing Figure 6, one can conclude that for the detection of aerosols in the atmosphere from wildfires, the most optimal heights are in the range of 10–20 m. Figure 7 shows experimental data on the concentration of aerosols with a particle diameter of at least 0.3 μm at a height of 20 m and 2 m at various distances from the fire source, as well as extrapolation curves. One can see from Figure 7 that with a combustion source size of 4 × 10 m, the maximum distance where the aerosol concentration in the air exceeds the background values at a height of 2 m and 20 m does not exceed 800 m and 2000 m, respectively. It is obvious that at a height of 20 m, based on the concentration of aerosol combustion products, it is possible to reliably record a smoke plume from a wildfire at a significantly greater distance.

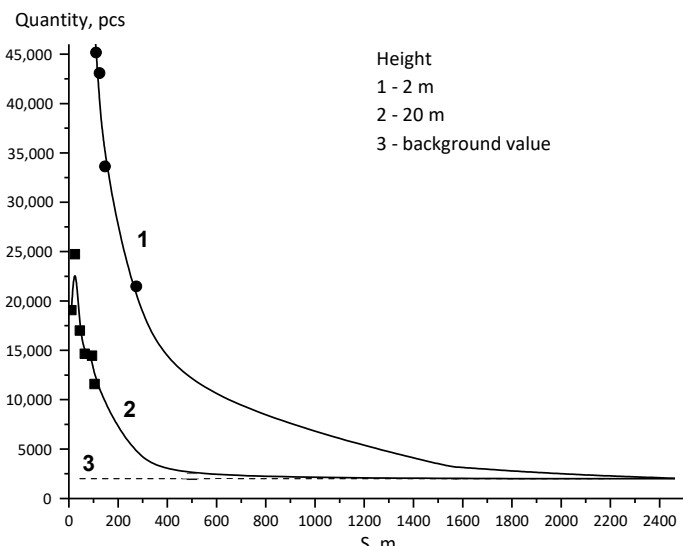

**Figure 7.** Concentration change of aerosols with a particle diameter of at least 0.3 microns at different heights in comparison with background value.

One can conclude from Figures 6 and 7 that a small source of wildfire in the wind direction leads to an increase in the concentration of condensed combustion products (plumes or aerosols) at a height of 20 m, exceeding the background values at distances up to 2000 m. It can be considered as a characteristic sign of a natural wildfire with regard to warning and operational signals about the source of ignition. In the case of a significantly larger wildfire source, associated with a larger mass of vegetation involved in combustion, the concentration of aerosols exceeding the background values will be recorded at much greater distances [36].

## 4. Conclusions

The presented results in this article clearly demonstrate that the impact of wildfires on atmospheric processes cannot be underestimated even while conducting an experiment with a model wildfire, the size of which is incommensurably smaller than the size of real wildfires, since the formation of atmospheric turbulence and the release of a significant amount of aerosols of small fractions that can be transported over considerable distances are observed. Obviously, atmospheric turbulence is formed as a result of heat release in the combustion zone and dissipation of turbulent structures in the flame. Considering the scale of real natural wildfires, one should understand that the formation of induced atmospheric turbulence will have a much larger scale, which will certainly affect global atmospheric processes. The release of gaseous and condensed combustion products and their transfer to higher layers of the atmosphere will affect not only the quality air and human health, but also global climate processes in general.

It is experimentally established that the values of the maximum values of $C_n^2$ obtained by optical and acoustic methods, as well as the maximum values of $C_T^2$ and $C_V^2$, in the case of a steppe fire [14] and a model crown fire have the same order. It is associated with the scale of turbulence in the flame and the dissipation of turbulent structures in it.

It can be concluded from the presented experimental data that aerosol components are emitted into the atmosphere as a result of a crown fire, the highest concentration of which corresponds to particle diameters of 0.3–0.5 μm. It is more than two times higher than the concentration of particles with diameters greater than 0.5 μm. Moreover, the highest concentration of these aerosols is observed at altitudes from 10 m to 20 m.

As a result of the experimental data extrapolation, one can assume that with the considered volume of burnt FVM (the experimental area size is 4 × 10 m), the excess concentration over the background value at a height of 20 m for small aerosol fractions will be observed at a distance of up to 2000 m from the combustion source. It is of interest



to further study these factors of the impact of wildfires on the atmosphere under the conditions of a real large natural wildfire and determine the limiting distance at which the concentration of aerosols will exceed the background values, depending on the size of the wildfire source.

**Author Contributions:** Conceptualization, E.L.; formal analysis, E.L., D.K., M.A. and V.R.; investigation, E.L., D.K., M.A., V.P., P.M., A.S., Y.L., A.L. and K.O.; methodology, E.L., D.K., V.P., M.A. and V.R.; project administration, D.K.; visualization, M.A., V.P., D.K., A.L. and E.L.; writing—original draft preparation, E.L., D.K. and V.R.; writing—review and editing, M.A., V.R., D.K. and E.L. All authors have read and agreed to the published version of the manuscript.

**Funding:** This work was supported by the Russian Science Foundation, grant Number 20-71-10068.

**Data Availability Statement:** The data presented in this study are available on request from the first and the corresponding author.

**Conflicts of Interest:** The authors declare no conflict of interest. The founding sponsors had no role in the design of the study; in the collection, analyses, or interpretation of data; in the writing of the manuscript; or in the decision to publish the results.

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
