# Peer review of "Crown Fire Modeling and Its Effect on Atmospheric Characteristics"

_atmosphere, doi:10.3390/atmos13121982_

Round 1

Reviewer 1 Report

This study investigates how wildfire affects the atmospheric turbulent structure and the aerosol concentrations over the canopy layer by experimental approaches in a semi-natural scenario. There are several concerns here.

1) this study has not addressed the significance of the examination of the atmospheric turbulent structure there, please add one more paragraph and references there.

2) equation 3, C_T is not expressed in a standard way.

3) line 152, 'heat release in the combustion zone and dissipation of turbulent structure in the flame'. Any physical interpretation of this and cite more references for comparison. This conclusion refers to what kind of implementation or application.

4) more sophisticated analyses beyond those exhibited in figures 5, 6, and 7 are highly recommended. But again these depend on what is the final goal of the significance of your targeted research.

Author Response

Dear Reviewer,

Thank you for your valuable comments and suggestions. I have presented below answers to questions: 

 (Point 1): This study has not addressed the significance of the examination of the atmospheric turbulent structure there, please add one more paragraph and references there.

Response 1: The authors agree with this remark. Information has been added to the introduction section.

(Point 2): equation 3, C_T is not expressed in a standard way.

Response 2: The equation 3 has been corrected.

(Point 3): line 152, 'heat release in the combustion zone and dissipation of turbulent structure in the flame'. Any physical interpretation of this and cite more references for comparison. This conclusion refers to what kind of implementation or application.

Response 3: We agree with the reviewer’s comments. The following information has been added to the article: “The magnitude of the structural characteristic of the air refractive index fluctuations is often used in problems of atmospheric optics for description of the turbulence effect on optical radiation. The work [Kovadlo P.G., Lukin V.P., Shikhovtsev A.Yu. Development of the model of turbulent atmosphere at the Large solar vacuum telescope site as applied to image adaptation // Atmos. Ocean. Opt. 2019. V. 32, N 2. P. 202–206.] shows justification for the fact that turbulence in the atmosphere surface layer has a decisive influence on the profile C_n^2 (z) up to heights of the several kilometers order (C_n^2 (z) is the vertical profile of the structural characteristic of air refractive index fluctuations).”

(Point 4): more sophisticated analyses beyond those exhibited in figures 5, 6, and 7 are highly recommended. But again these depend on what is the final goal of the significance of your targeted research.

Response 4: The authors are grateful to the reviewer for valuable comments, which undoubtedly help to improve the manuscript. The authors are planning to continue the development of this study, and the results will make it possible to draw significant practical conclusions and recommendations.

Reviewer 2 Report

Please see my comments as attached.

Author Response

Dear Reviewer,

Thank you for your valuable comments and suggestions. I have presented below answers to questions: 

(Point 1): In the abstract, line 17, what does a "lower layer of the forest model" mean? No enough background was given.

Response 1: We agree with the reviewer’s comments. There was an error in the translation. It was a "modeled" forest canopy in the experiment. Corrections have been added to the abstract.

(Point 2): In the abstract, please emphasis more on the innovations and significances of the current study. What recommendations can be made based on the current study?

Response 2: The abstract has been revised in accordance with the publisher requirements.

(Point 3): The introduction/literature review part can be further extended. What experimental and numerical approach has been adopted by researchers in the past and what advantages have been adopted in the current study?

Response 3: The authors are thankful for this remark. Information has been added to the paper.

(Point 4): monitoring and recording equipment at location 3? Please provide more details.

Response 4: Information has been added to the paper.

(Point 5): Figure 2, please provide more details on the experimental setup, e.g., dimensions of hr1, hr2, Hb etc. and what is the HRR of the fire?

Response 5: We agree with the reviewer’s comments. Information has been added to the article.

Determining HRR has not been the objective of our investigation. The heat release rate (HRR) of trees is certainly the factor than can contribute to fire spread in WUI fires. Little data exists in the open literature regarding the HRR of different tree species.

According [S.L. Manzello et al. Measurement of Firebrand Production and Heat Release Rate (HRR) from Burning Korean Pine Trees] using the Babrauskas correlation results in a calculated peak HRR of 2.8 MW and 12.2 MW for the 2.4 m crown height and 4.5 crown height Douglas-Fir trees, respectively.  

(Point 6): Figure 5, please keep the vertical axis consistent by using 10-x instead of E-x.

Response 6: Corrections has added to the article.

(Point 7): Comparing the data shown in Figure 5 with the data published in [14] be beneficial if the comparisons are made by plotting the data on the same diagram.

Response 7: This is a valid point. We have added the comparison and revised figure 5b, 5c.

(Point 8): Figure 6 and 7, please add legends to the graph.

Response 8: We agree with the reviewer’s comments. The legends have been added.

Reviewer 3 Report

Very good relevance for the journal and technical quality.

Fair contribution on the field, good depth of research.

The abstract is clear including the specific objective of the work, the techniques employed and the significant results. There are too much words (248 words).

Good paper presentation and innovation.

The linguistic level and the mechanics of writing are appropriate for publication.

The references and quotations are clear and the bibliography is updated and relevant.

The conclusions are important, correctly explained and avoid misinterpretation. There are presented future plans.

The article is well organized. The introduction highlights current concerns in the research area and highlights the objectives of the study. The methods are clearly explained. The article has a logical structure and presents an adequate methodology.

The title clearly and concisely expresses the content of the article.

The presentation is clear, the figures, tables and equations are visible.
The tables correctly indicate the units of measurement, and the data presented in the tables are correctly evaluated and interpreted in the article.
The graphs and figures correctly illustrate the topic discussed, and the data presented in the graphs are correctly evaluated and interpreted in the article. The graphics and figures are well proportioned and aesthetically placed in the article.

Accept this paper with minor revisions about the lenght of the abstract and adding the Discussion section. Please revise the References section about the format according with the template.

Author Response

Dear Reviewer,

Thank you for your valuable comments and suggestions. I have presented below answers to questions:

(Point 1): Accept this paper with minor revisions about the length of the abstract and adding the Discussion section.

Response 1: Thank you for your positive evaluation of our work

(Point 2): Please revise the References section about the format according with the template.

Response 2: We agree with the reviewer’s comments. Corrections has been added to the article.

Round 2

Reviewer 1 Report

Accepted in present form.

Reviewer 2 Report

The authors have addressed all my comments properly. The manuscript can be accepted in its current form.